# AFM Images of Viroid-Sized Rings That Self-Assemble from Mononucleotides through Wet–Dry Cycling: Implications for the Origin of Life

**DOI:** 10.3390/life10120321

**Published:** 2020-11-30

**Authors:** Tue Hassenkam, Bruce Damer, Gabriel Mednick, David Deamer

**Affiliations:** 1Globe Institute, University of Copenhagen, DK-1350 Copenhagen, Denmark; 2Department of Biomolecular Engineering, University of California, Santa Cruz, CA 95064, USA; bdamer@digitalspace.com; 3UpRNA, 250 Natural Bridges Drive, Santa Cruz, CA 95060, USA; gmednick@gmail.com; 4Department of Chemistry and Biochemistry, University of California, Santa Cruz, CA 95064, USA

**Keywords:** origin of life, RNA, wet-dry cycle, AFM, viroid

## Abstract

It is possible that early life relied on RNA polymers that served as ribozyme-like catalysts and for storing genetic information. The source of such polymers is uncertain, but previous investigations reported that wet–dry cycles simulating prebiotic hot springs provide sufficient energy to drive condensation reactions of mononucleotides to form oligomers. The aim of the study reported here was to visualize the products by atomic force microscopy. In addition to globular oligomers, ring-like structures ranging from 10–200 nm in diameter, with an average around 30–40 nm, were abundant, particularly when nucleotides capable of base pairing were present. The thickness of the rings was consistent with single stranded products, but some had thicknesses indicating base pair stacking. Others had more complex structures in the form of short polymer attachments and pairing of rings. These observations suggest the possibility that base-pairing may promote polymerization during wet–dry cycling followed by solvation of the rings. We conclude that RNA-like rings and structures could have been synthesized non-enzymatically on the prebiotic Earth, with sizes sufficient to fold into ribozymes and genetic molecules required for life to begin.

## 1. Introduction

Our current understanding of the early Earth suggests that life originated approximately four billion years ago [1,2,3], and that early stages of life depended on polymers resembling RNA [4,5]. It has been proposed that organic compounds from geochemical sources and meteoritic infall [6] likely accumulated on volcanic land masses emerging from a global ocean [7]. Some were flushed by precipitation into hydrothermal pools supplied with fresh water distilled from a salty ocean [8,9]. Water levels in some pools fluctuated due to variations in the flow of hot spring water, geyser activity and precipitation. Solutes would be concentrated by evaporation and form films on hot mineral surfaces in which condensation reactions synthesize biologically relevant polymers [8,9,10]. For instance, it has been shown that short oligomers linked by peptide bonds are synthesized from mixtures of amino acids during wet–dry cycling [11,12,13]. More recently it has been reported that ribonucleotides can be synthesized from simple organic compounds in conditions that incorporate a wet-dry cycle [14,15].

If so, nucleotide monomers present as solutes in the pools may also have formed polymers under these conditions on the prebiotic Earth. In previous studies, we have reported a series of experimental results supporting the conclusion that such conditions can drive polymerization of mononucleotides:Oligomers produced by wet-dry cycling of mononucleotides form pellets when isolated by precipitation in 70% ethanol or with spin tubes designed to purify nucleic acids. The oligomers exhibit UV spectra that are characteristic of the mononucleotides composing them.The RNA-like oligomers synthesized by wet–dry cycles are recognized by the enzymes used to label RNA with radioactive phosphate. When the labeled material is analyzed by standard methods of gel electrophoresis, it moves through the gel as expected for polyanions and shows up as a band ranging from 20 to >100 nucleotides in length [16]. The oligomers can also bind dyes such as ethidium bromide and SYBR-SAFE that are used to stain RNA polymers in gels [17].When tested by nanopore analysis with the alpha-hemolysin pore, the oligomers produce blockade signals virtually identical to those caused by single stranded RNA molecules. This demonstrates that at least some of the products are linear polyanionic strands that impede ionic currents as they are driven through the nanopore by an applied voltage [16,17,18].If a 1:1 mole ratio of AMP and UMP is exposed to wet–dry cycling, the oligomeric products exhibit hyperchromicity, but the products from AMP alone do not [17]. This result is consistent with hairpin structures forming in random sequence linear polymers of RNA composed of monomers capable of Watson-Crick base pairing.An X-ray diffraction study of AMP-UMP mixtures revealed that linear arrays of stacked bases are present with 3.4 Å distances between the bases [19]. The arrays are referred to as pre-polymers which presumably can form phosphodiester bonds during wet–dry cycles that link them into polymers by condensation reactions.We also note that cyclic guanosine monophosphate has been shown to undergo spontaneous polymerization in the dry state resulting in oligomeric products up to 80 nucleotides in length [20].

Although the experimental evidence is consistent with oligomers being formed during wet-dry cycling [10,16,17,18,21], the structure of the oligomers is unknown. We therefore employed atomic force microscopy (AFM) to visualize the products at single molecule resolution.

## 2. Materials and Methods

We exposed dilute solutions of mononucleotides such as adenylic (AMP), uridylic (UMP), guanylic (GMP) and cytidylic (CMP) acid to conditions simulating hot hydrothermal pools undergoing wet–dry cycles. For convenience, hereafter the mononucleotides will sometimes be abbreviated A, U, G and C, and mixtures will be abbreviated AU and GC for 1:1 mole ratios of AMP:UMP or GMP:CMP, respectively.

A single cycle is defined here as a 10 mM solution of mononucleotides evaporating on mica (20–40 µL) or glass (100 µL) at 80 °C for 30 min, then rehydrated by addition of the same volume of water. Because the nucleotides were acids rather than sodium salts, the pH was approximately 2.5. A typical experiment included three such cycles followed by flushing with water to dissolve and remove excess mononucleotides. The mica was then dried for AFM examination.

### 2.1. Atomic Force Microscopy

We used a Cypher from Asylum (now Oxford instruments) equipped with a standard AC240 silicon tip from Olympus with a spring constant around 2 nN/nm and a resonant frequency around 70 kHz. Images were at least 512 × 512 pixels and aimed for a resolution of 1 nm per pixel or less. The actual resolution is determined by several factors. The combination of a 0.1 Å resolution in the z-direction combined with a radius of curvature of the tip between 5 and 10 nm, puts the highest achievable resolution to 2–3 Å in the x,y direction. Meaning that two nucleotides would have to be separated by more than 2–3 Å for the AFM to be able detect two individual molecules. In our current setup the gap would have to be more than about 10 Å for us to be able to detect it. The scanning was performed in ambient conditions at 1 Hz. The height of the rings (thickness) was measured using the section analysis tool in the Igor pro control software for the AFM.

### 2.2. Compounds and Materials

Sources of mononucleotides: adenosine 5’-monophosphate (US Biochemical, Thermo Fisher Waltham, MA, USA), uridine 5’-monophosphate (P-L Biochemicals, Milwaukee, WI, USA), guanosine 5’ monophosphate (MP Biochemicals, Irvine, CA, USA), cytidine 5’-monophosphate (Sigma-Aldrich St. Louis, MO, USA). The adenosine and uridine, or cytidine and guanosine nucleosides were purchased from Sigma-Aldrich. We used ultra-pure deionized water (Milli-Q, St. Louis, MO, USA) from a Millipore system to prepare 10 mM solutions of the nucleotides.

Mica was purchased from SPI Supplies (West Chester, PA, USA) (Mica grade v-4). We used specially designed glass slides having three small Scotch tape was used to repeatedly cleave the mica until a complete cleavage was achieved (usually 3–5 times). wells (1.5 cm in diameter and 0.5 mm deep) to run the wet–dry cycles on glass. Prior to use, the glass slides were cleaned in tap water, ethanol, ultra-pure deionized Milli-Q, then exposed for 40 min to ozone cleaning in a UV/Ozone procleaner from Bioforce nanosciences (Chicago, IL, USA). In addition to the controls described in the text, we ran blind tests with wet–dry cycles using only (Milli-Q) water to verify the cleaning procedures. This assured that the rings were not created by an unknown contaminant in the water.

### 2.3. Wet-Dry Cycles on Glass Substrates

The initial experiments used acid forms of two mononucleotides, adenylic acid (AMP) and uridylic acid (UMP) in 1:1 mole ratios. We chose mononucleotides capable of base pairing, expecting that they might produce more stable polymers than single nucleotides. The acid forms of the nucleotides lowered the pH to 2.5 as expected for a 10 mM solution of a weak acid. This pH range simulates the acidity of water circulating in hot springs and favors condensation reactions that link mononucleotides by phosphoester bonds. Experiments with individual mononucleotides were performed as controls. In preliminary experiments, the reactions were carried out on microscope slides having three wells into which 0.1 mL of 10 mM nucleotide solutions were added. The slides were placed on a laboratory hot plate set to 80 °C. The solution evaporated in a few minutes, then remained in a dry state for 30 min during which phosphoester bonds were expected to form by condensation. Because none of the components could be readily oxidized by molecular oxygen, the experiments were carried out in open air.

After 30 min, each well was rehydrated with 50 µL of water and the reactants were stirred for a few seconds with the end of a cleaned stainless steel spatula. At the end of three such wet-dry cycles 100 µL of water was added and transferred from one well to the next to dissolve any products. This was repeated two more times and each 100 µL sample was placed in a 1.5 mL Eppendorf centrifuge tube. Half of the 300 µL total was treated by a standard ethanol procedure used to precipitate oligonucleotides as pellets which were then dissolved in 50 µL of water. Yields varied from one run to the next but typically amounted to tens of micrograms, representing up to 10% yields of polymers based on the monomers present as reactants. 

To determine whether rings were present, a 20 µL aliquot of the cycled reaction mixture with products was placed on freshly cleaved mica for 30 s to allow polymers to adhere, then flushed with ultra-pure deionized water (Milli-Q). The surface was dried under a stream of nitrogen gas, then examined by atomic force microscopy. This method was developed for studying DNA [22].

### 2.4. Experiments on Mica Surfaces

We added 20–40 µL of the mononucleotide solutions to the surface of freshly cleaved mica which was cut to fit the stage of the AFM. The mica with nucleotide solution was then placed on a laboratory hot plate maintained at 80 °C. The solution was allowed to dry for 30 min after which the mica surface was rehydrated with 20–40 µL ultra-pure deionized water (Milli-Q). This was repeated 3 times. After the final dry cycle, the mica surface was flushed with ample amounts of water (Milli-Q) (3–5 mL) and dried under a gentle stream of nitrogen gas. The sample was immediately mounted and examined in the AFM.

## 3. Results and Discussion

Figure 1a,b shows a typical result from an AU solution sampled after the last of three wet-dry cycles at 80 °C on a glass substrate. The AFM images are shown in inverted height scale, from light blue to black. We expected to see products appear as small particles of oligomers that had folded into globular forms. Such particles were abundant, but we also observed multiple ring-like structures. The largest was approximately 200 nm in (outer) diameter while the smallest detectable ring was approximately 10 nm in diameter. The average diameter over 100 rings was 39 ± 19 nm (more rings are shown in Appendix A). Smaller rings might have been present in large numbers but due to convolution caused by the tip shape the AFM could not distinguish between a solid particle and rings less than around 10 nm in diameter. The rings shown in Figure 1c,d differ from the rings in Figure 1a,b because the wet dry cycle was performed on a freshly cleaved mica surface rather than glass, and the nucleotides were a mixture of GMP and CMP instead of AMP and UMP.

The results suggest that rings were produced only when complementary base pairs are present in the solution. The AU and GC rings were similar in thickness, and with similar variations in the integer values of the single layer (cross sections in Figure 2). While most rings were a single layer thick (0.3–0.4 nm), parts of the rings were often multiple layers in thickness, suggesting a packing motif with monomers building sequential layers of rings. Multiple examples are found in Figure 1, in which pairs of rings having the same diameter but with twice the thickness are indicated by red arrowheads.

Figure 2 shows further examples at a higher magnification. In Figure 2a, a cross section is drawn across a single layer ring and a ring with partial double layer. In Figure 2b, a cross section is drawn across a full double ring AU and a full double GC ring is shown in Figure 2d. In Figure 2c, the cross section is drawn across a ring with a particle attached to the ring. These types of rings that came decorated with extra material were abundant. There were also many rings that appeared to have small tails of polymeric segments, partial extra layers and small particles attached. Some rings appeared to form pairs either with smaller rings attached to larger rings, or by similar sized rings tethered by short polymer strands. A catalog of these different types rings and structures is illustrated in Figure 3.

The observed diameter of the GC rings did not vary as much as the AU rings, with a size distribution ranging from 10 to 60 nm and an average diameter of around 30 ± 7 nm for 1000 rings found in a 12.5 µm^2^ area. (Figure 1 and Appendix A).

We also ran experiments with individual nucleotide solutions to see whether they generated rings (Figure 4). We observed a few particles in the expected size range, but no rings. Furthermore, the number of particles was far less than the number of observed rings in a given area. We concluded that nucleotide mixtures capable of base pairing were required for ring formation.

The rings produced from the AU base pairs displayed a larger distribution of sizes although the average diameter was close to that of the GC rings: 39 ± 19 nm. The AU sample was taken from a solution; however, so it probed a mixture of whatever had dissolved in water afterwards followed by adsorption to the mica surface. Perhaps the size differences were simply due to variations between rings being first desorbed from glass and then adsorbed to mica and; therefore, rings from the entire surface of the glass substrate were sampled, compared to the GC mixture cycled on mica where only small regions could be probed.

While the AFM images are consistent with single molecule rings, we cannot, based on these images, rule out small gaps between individual nucleotides (AFM resolution described in the methods section). However, we find it entropically unlikely that the individual nucleotides would align in rings of about the same size and only when there was matching base pairs, unless the alignment was accompanied by some binding between molecules. Since we only observe the rings with mixtures of complementary nucleotides, one possibility is that small linear polymers are formed at first that are partially double stranded. Some of these have ends that can undergo base pairing which then closes the ring and allows a linking ester bond to form. This process is consistent with the apparent double layers and the non-circular loops we observed, together with attachments and more complex structures shown in Figure 3 (rows 3 and 4).

The number of nucleotides present in the rings assuming a single stranded molecule can be estimated from their diameters (10–200 nm) corresponding to 30–600 nm in circumference. The number of bases in the rings would then be 50–1200 if we use 0.5 nm as the spacing of nucleotides in a single strand, or 70–1760 base pairs if the rings are duplex strands having 0.34 nm spacing. The average length found in gels was 50 nucleotides, with a range from 20–100 nucleotides [16,17]. These presumably produce the small particles that represent the major fraction observed in the AFM images. The rings are likely to be rare and the big ones are even rarer. In AFM the rings stand out because they are easily identifiable and, because of their size, they might adhere more readily to mica, but in reality they are a small fraction of the polymers present in the mixture.

We performed several control experiments to verify that the rings were indeed nucleic acid polymers bound by phosphodiester bonds and not random contamination or artifacts. For instance, we confirmed the presence of rings in two completely independent sets of experiments performed in two different laboratories and repeated the experiments with fresh compounds, solutions, and new tips and surfaces. We tested the mononucleotide solutions to assure that they did not contain contaminating rings before exposure to wet–dry cycles (Appendix A). We ran wet–dry cycles with individual mononucleotides to see if they also generated rings (Figure 4). Structures consistent with oligomers can be observed in all the images, but rings like the ones shown in Figure 1 and Figure 2 were absent. We also ran wet–dry cycles using two different water sources without mononucleotides and verified that the solvents did not contain compounds that could generate rings.

Similar to the control with mononucleotides, we conducted a second control with nucleosides to test whether the rings were simply drying artifacts produced by matching base pairs. To this end, we performed wet–dry cycles with matching pairs of nucleosides that lacked phosphate groups. If nucleosides also produced multiple rings, that would argue against the presumed polymerization of nucleotides. We exposed adenosine and uridine, or cytidine and guanosine nucleoside mixtures to wet–dry cycling on mica. We observed a few disk-shaped structures with diameters similar to the rings but these lacked the distinctive ring-shaped perimeters (Figure 5). The disk shape and other structural features are evident from the accompanying AFM amplitude images (Figure 5b,d).

Controls were also run with uncycled 10 mM mononucleotides on mica. The protocol was the same as that used in Figure 1a,b and Appendix A. A few microliters of solution were added to a freshly cleaved mica surface, left there for 30 s and then rinsed with pure water. Only particles were apparent, showing that the solutions were not by themselves sources of rings (Appendix A).

In another set of controls, AU and GC were wet–dry cycled at room temperature on a mica substrate. Although none of the controls displayed rings like those observed in Figure 1 and Figure 2, the GC mixture dried at room temperature did self-assemble into rod-like structures (Appendix A) that have been reported previously [23]. It is well known that GMP spontaneously forms tetrameric quadraplexes which assemble into linear rods. Base stacking has also been observed by X-ray diffraction when AU mixtures are dried on various substrates [19]. These are referred to as pre-polymers and may precede polymerization in wet–dry cycling. Rings are created when AU solutions undergo wet–dry cycling on glass surfaces, so mica is not a specific catalytic surface, but we have not excluded the possibility that the slightly negative SiO groups, on the silicate surfaces of glass and mica, could play a role in the polymerization as has been previously suggested [24,25].

In summary, AFM images of polymeric products synthesized by three wet–dry cycles of mononucleotides revealed numerous particulate structures, as expected if linear fragments of short oligomers folded into globular conformations that adhered to the mica surface. If base-pairing nucleotides were present in the mixture, such as AMP + UMP or GMP + CMP, obvious ring-like structures were also apparent. Polymerization related to ester bond synthesis is thermodynamically favored in wet–dry cycling that simulates conditions present on the prebiotic Earth. If the polymers grow to sufficient lengths during cycles, it seems plausible that some of them will happen to join their ends and form the rings observed here. The ring structures add to the weight of evidence that the oligomers synthesized by wet–dry cycles resemble RNA in their physical properties [26]. Furthermore, the sizes of the rings are well beyond the minimum range required for catalytic or genetic functions. In addition, similar ring structures has been observed with biological RNA and DNA [27,28]. From their diameter and estimates of nucleotide packing in nucleic acids, it is possible to calculate that up to several hundred nucleotides compose the larger rings. This is in the same range as the number of nucleotides in viroids, the smallest known infective agents composed of RNA rings. Viroids were discovered in 1971 by Theodor Diener, who speculated later that they may represent RNA remnants of an early form of life [28].

A simple calculation shows that a few micrograms of RNA-like polymers synthesized by wet–dry cycling represent trillions of molecules, each different from all the rest in length and nucleotide sequence. A goal of future research will be to test whether populations of such ring-like polymers can undergo selection in response to environmental conditions. For instance, the nucleotide composition or sequences of some rings might make them increasingly stable to spontaneous hydrolysis. If so, it would represent the first instance of Darwinian evolution and an early step toward the origin of life as we know it. In a letter to his colleague Joseph Hooker, Charles Darwin speculated that life might begin in a “warm little pond” [29]. The results reported here are consistent with Darwin’s suggestion, with the proviso that the pond must be an acidic solution of monomers undergoing wet–dry cycling at elevated temperatures.

## Figures and Tables

**Figure 1 life-10-00321-f001:**
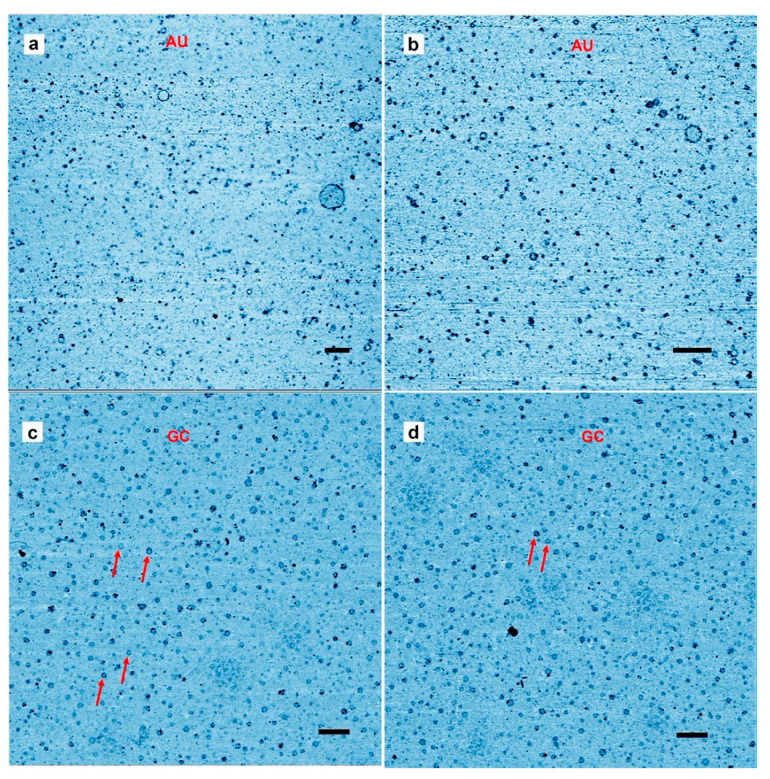
Representative Atomic force microscopy images of rings. (**a,b**) Rings captured on a mica surface from a solution containing polymeric products after three wet–dry cycles of an adenylic, and uridylic acid (AU) solution on a glass surface. The mica surface was exposed to the solution for about 30 s as described in the methods section. (**c**,**d**) the mica substrate used for hot wet-dry cycles with a guanylic and cytidylic acid (GC) solution. Scale bars shows 200 nm.

**Figure 2 life-10-00321-f002:**
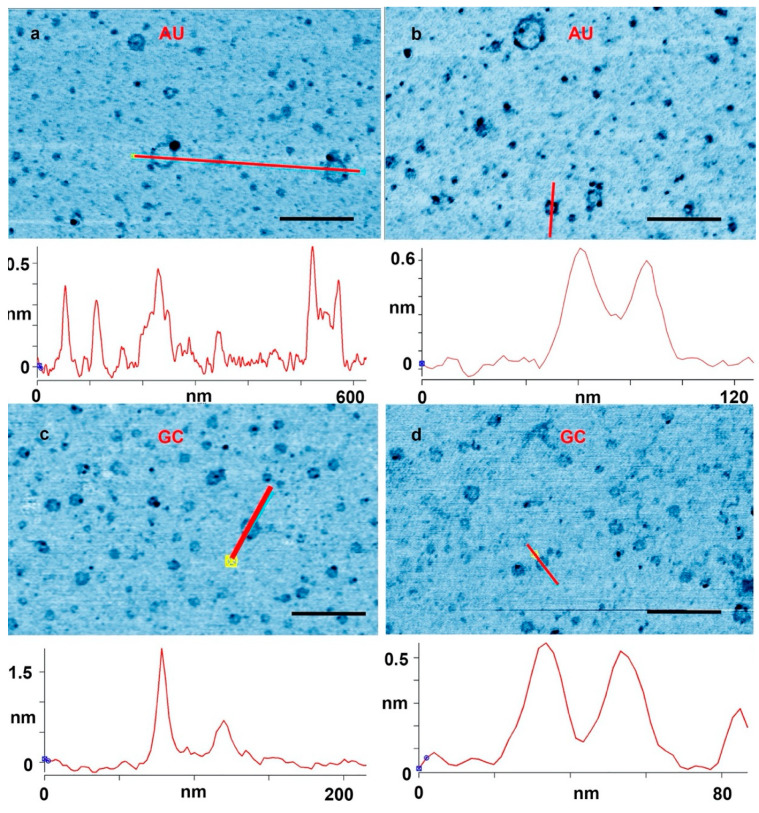
(**a**,**b**) Rings captured on a mica surface from an adenylic and uridylic acid (AU) solution after three wet–dry cycles on glass. (**c**,**d**) A mica substrate used for three wet–dry cycles with a guanlylic and cytidylic acid (GC) solution. The cross sections were drawn along the red lines marked in the images. Scale bars shows 200 nm.

**Figure 3 life-10-00321-f003:**
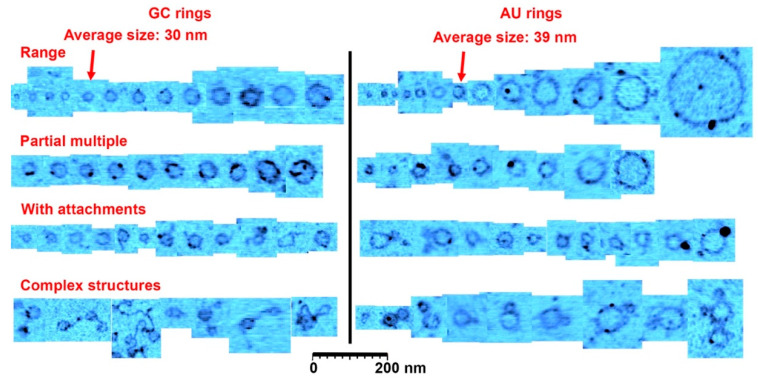
A catalog of ring types. This figure shows a representative range of ring sizes found on mica samples after wet-dry cycling of guanylic and cytidylic acid (GC) solutions and adenylic and uridylic acid (AU) solutions. Most rings were the size of the average indicated with the arrowheads. The second row shows rings that had segments twice or more in thickness (there is also examples in the other rows), and the third row shows rings with small polymer or particle attachments. The last row shows rare complex structures observed in the images, very often involving several rings attached to each other. The cutouts showing rings and structures were extracted from approximately 30 atomic force microscopy images to reflect the range of structures found. Scale bar shows 200 nm.

**Figure 4 life-10-00321-f004:**
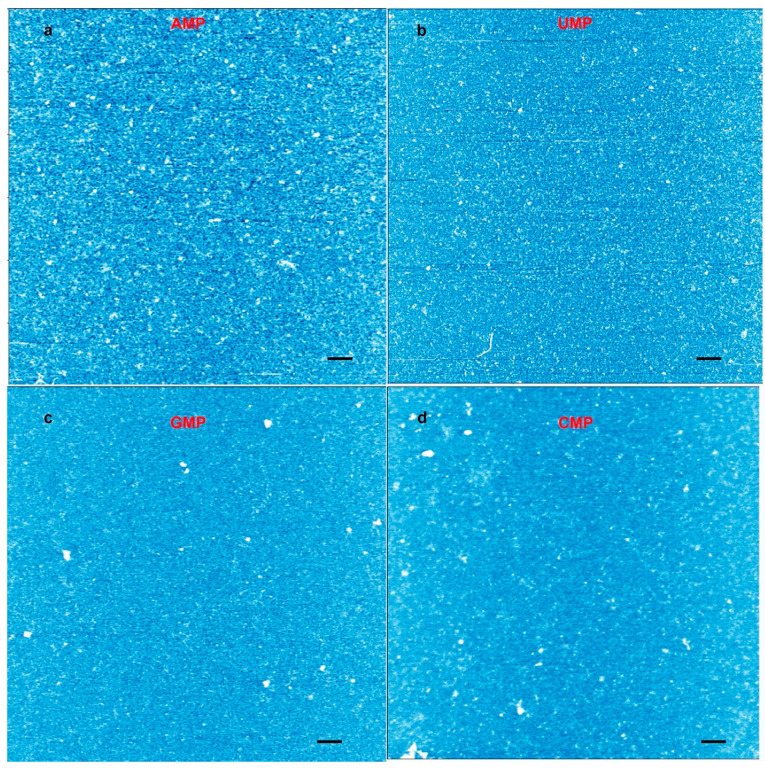
Atomic force microscopy (AFM) images from a control experiment performed with individual mononucleotide solutions cycled three times on mica. (**a**) AFM image of an adenylic acid solution cycled on mica; (**b**) uridylic acid; (**c**) cytidylic acid; and (**d**) cytidylic acid. The height scale runs from light blue (low) to white (high). The images represent images obtained from eight to 10 different spots across the mica surface. No rings were observed. The scale bars show 200 nm.

**Figure 5 life-10-00321-f005:**
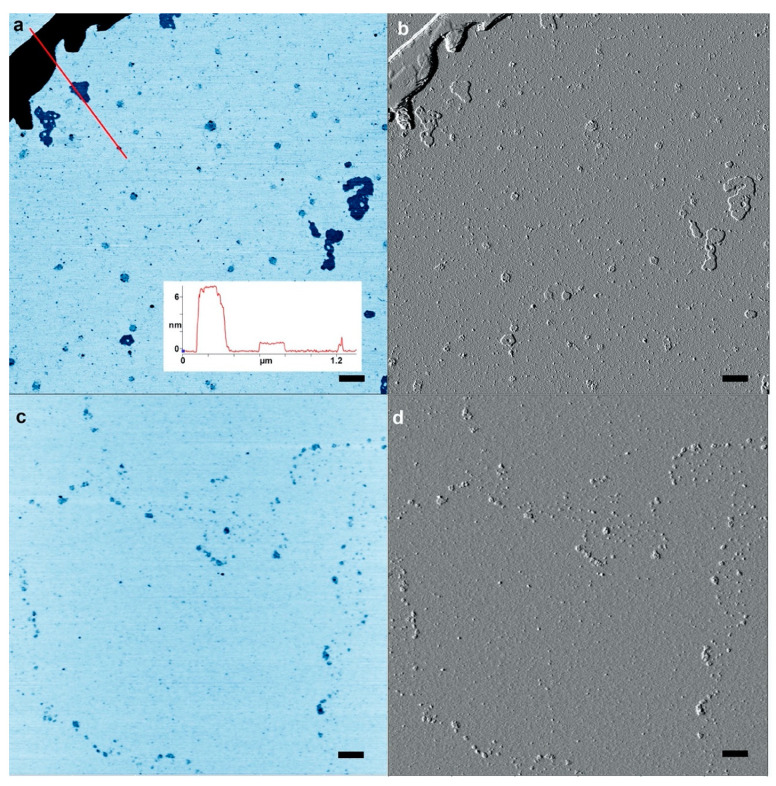
A control experiment was performed with matching pairs of nucleoside mixtures cycled three times on mica. (**a**) Atomic force microscopy (AFM) image of a 1:1 adenosine:uridine mixture inverted for contrast. (**b**) AFM image of the area shown in (a) but using AFM amplitude mode to reveal structures. (**c**,**d**) the same image pairs with a 1:1 mixture of guanosine and cytidine. The scale bars show 200 nm.

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
