# Peer review of "AFM Images of Viroid-Sized Rings That Self-Assemble from Mononucleotides through Wet–Dry Cycling: Implications for the Origin of Life"

_life, 2020, doi:10.3390/life10120321_

Round 1
Reviewer 1 Report
wet dry cycles simulating hot springs may drive condensation reactions of mononucleotides to form oligomers, but do not explain interactional motifs of RNA stem loop consortia with which biological selection started
Author Response
It is beyond the scope of the paper and would be too speculative to go into the discussion of selection. We simply wanted to show the products from a wet-dry cycle with RNA nucleotides. To our surprise we saw a lot of rings. We suggest that the basic requirements for Darwinian evolution are now in place, with structural units that can contain base sequences with the potential to contain genetic information. We have added a sentence to clarify what we think the paper is showing.
Reviewer 2 Report
This paper is fully speculative. The conclusions are not supported by presented results.
The structures observed by use of AFM are not shown to be polynucleotides, and it remains unknown what these structures are. If the authors state that these structures represent linear and circular polynucleotides, evidence for that should be presented before the paper can be considered for publication. I would suggest NGS protocols designed for short RNAs to clarify the point.
Author Response
How is the paper fully speculative? We present experimental results from atomic force microscopy, not speculations. We offer an interpretation of the results which is consistent with our previous studies of nucleotide polymerization. We only observe structures when we use dilute solutions of nucleotides that are allowed to evaporate on mica surfaces, or exposed to solutions from wet-dry cycles on glass and which subsequently were brought into contact with mica surfaces. There is nothing present in the solutions except nucleotides and pure water. In our previous studies on similar systems we have confirmed the presence of RNA oligomers and polymers and used multiple techniques to characterize the products. We described this in the introduction and cited the earlier published studies. The manuscript builds on a foundation of previous research. The results we report here simply reveal the structures of the polymers, and we were surprised to see that some of them were in the form of rings.
Reviewer 3 Report
I find fhis article by David Diemer and colleagues be very comprehensive and interesting. This work can be published without changes.
Author Response
We thank the referee for understanding the significance of the results we presented.
Reviewer 4 Report
The authors address a long-standing challenge: the synthesis of long RNA molecules under prebiotically relevant conditions. The present manuscript claims that large rings are formed from nucleotides upon wet-dry cycling. The evidence rests solely on analysis by AFM, which constitutes a critical weakness of the work. AFM is a technique that is prone to artefacts that arise upon evaporating liquids on a surface. While some controls are presented that address this issue, confirmation of the structures formed through another technique (like TEM and DLS) is needed. What is also missing in the present paper is a quantitative analysis. If large RNA molecules indeed exist, then what fraction of the total amount of material do they represent (why are they not visible on the gels?)? And how did they get so large? Answers to these questions are relevant to allow the reader to assess to what extent the present work contributes to the challenge to access large RNA molecules from nucleotides.
At present the evidence is far too limited and much more thorough and extensive analytical efforts are needed before I can support publication of this work.
Author Response
We have shown the original comments by the reviewer in plain text and our response in bold
The authors address a long-standing challenge: the synthesis of long RNA molecules under prebiotically relevant conditions. The present manuscript claims that large rings are formed from nucleotides upon wet-dry cycling. The evidence rests solely on analysis by AFM, which constitutes a critical weakness of the work. AFM is a technique that is prone to artefacts that arise upon evaporating liquids on a surface.
While some controls are presented that address this issue, confirmation of the structures formed through another technique (like TEM and DLS) is needed.
We do not only see the rings when we dry solutions on the surface. In other cases we expose a surface to water, but after short period we blast the water away with a puff of nitrogen gas. So even using completely different procedures for removing the water we see the same type of rings. We have written this in the manuscript.
We agree that AFM images should be analyzed with careful attention to certain details, and we described how we avoided artifacts in the methods section by using different AFM tips and reproducing the result with a new sets of tips and surfaces. When analyzing the width of the observed rings we also account for tip shape. But to dismiss the images just because they were made by AFM is to dismiss the whole AFM technique.
We don't see how TEM and DLS would add anything new. Doing direct HRTEM on RNA/DNA is not a trivial issue, and was only achieved recently.
Furthermore, we do not rely solely on AFM analysis. The AFM results build on a foundation of previous studies in which we demonstrated the presence of nucleotide polymers after wet-dry cycles. We have cited these studies in the manuscript. The AFM results are supported by a series of controls which seemed to have been overlooked.
What is also missing in the present paper is a quantitative analysis. If large RNA molecules indeed exist, then what fraction of the total amount of material do they represent (why are they not visible on the gels?)?
In our previous work, we reported that each wet-dry cycle produces yields of a few percent of polymers that can be isolated by standard ethanol precipitation methods. The polymers do show up in gels, both by 32-P labeling and by interactions with fluorescent dyes. The estimated lengths range from 20mers to >100mers, and we assume that the rings are present along with linear polymers.
In the current study we did present quantitative data that can be extracted from the AFM images, including height, ring diameters and average size. These results are consistent with other investigations that used AFM to examine single and double stranded nucleic acids.
And how did they get so large?
This question will be addressed in our future studies. However, we have previously shown that polymers can exceed 100 nucleotides in length after multiple wet-dry cycles. When chain lengths become very long, the ends can join by forming phosphoester bonds, the same reaction by which they add nucleotides to grow in length. It is known that biological RNA often forms ring structures that resist hydrolysis.
Answers to these questions are relevant to allow the reader to assess to what extent the present work contributes to the challenge to access large RNA molecules from nucleotides.
If readers look into our earlier published studies they will find answers to many of the questions posed by this referee.
At present the evidence is far too limited and much more thorough and extensive analytical efforts are needed before I can support publication of this work.
Once again, we have cited our previous studies that cover what this referee is asking for, showing that oligomers and polymers are produced in wet-dry cycles. All we are doing here is to report that some of them are in the form of rings. We believe this has profound implications that should be shared with the community.
Round 2
Reviewer 4 Report
The concerns I had in my previous assessment remain. The authors have not done much to address these. The results are potentially significant but more solid evidence, based on complementary techniques is required.
Author Response
We are sorry we could not satisfy the reviewer. We have changed the text to underline the limitations of the AFM technique. However we respectfully disagree that we have not used other techniques to verify the presence of polymers. The AFM merely display the shape of these, and we are convinced that this is the best technique for this particular task.